# Abundance of Selected Lipopolysaccharide-Rich Bacteria and Levels of Toll-like Receptor 4 and Interleukin 8 Expression Are Significantly Associated with Live Attenuated Rotavirus Vaccine Shedding among South African Infants

**DOI:** 10.3390/vaccines12030273

**Published:** 2024-03-05

**Authors:** Lerato P. Kgosana, Mapaseka L. Seheri, Cliff A. Magwira

**Affiliations:** 1Diarrheal Pathogens Research Unit (DPRU), Department of Medical Virology, Sefako Makgatho Health Sciences University, Pretoria 0208, South Africa; lerato.kgosana@smu.ac.za (L.P.K.); mapaseka.seheri@smu.ac.za (M.L.S.); 2Department of Medical Virology, School of Medicine, Sefako Makgatho Health Sciences University, Ga-, Pretoria 0208, South Africa

**Keywords:** rotavirus, rotavirus vaccine shedding, bacterial lipopolysaccharide, Toll-like receptor 4, interleukin-8, vaccine adjuvant

## Abstract

Bacterial lipopolysaccharides (LPSs) have been shown to promote enteric viral infections. This study tested the hypothesis that elevated levels of bacterial LPS improve oral rotavirus vaccine replication in South African infants. Stool samples were collected from infants a week after rotavirus vaccination to identify vaccine virus shedders (*n* = 43) and non-shedders (*n* = 35). Quantitative real-time PCR was used to assay for selected LPS-rich bacteria, including *Serratia marcescens*, *Pseudomonas aeruguinosa* and *Klebsiella pneumonia*, and to measure the gene expression of bacterial LPS, host Toll-like Receptor 4 (TLR4) and Interleukin-8 (IL-8). The abundance of selected LPS-rich bacteria was significantly higher in vaccine shedders (median log 4.89 CFU/g, IQR 2.84) compared to non-shedders (median log 3.13 CFU/g, IQR 2.74), *p* = 0.006. The TLR4 and IL-8 gene expressions were increased four- and two-fold, respectively, in vaccine shedders versus non-shedders, but no difference was observed in the bacterial LPS expression, *p* = 0.09. A regression analysis indicated a significant association between the abundance of selected LPS-rich bacteria and vaccine virus shedding (Odds ratio 1.5, 95% CI = 1.10–1.89), *p* = 0.002. The findings suggest that harbouring higher counts of LPS-rich bacteria can increase the oral rotavirus vaccine take in infants.

## 1. Background

Rotavirus remains one of the prominent causes of childhood morbidity and mortality, particularly in lower-income countries [1]. Vaccines for preventing severe diarrhoea due to rotavirus infection are available but do not perform as well in lower-income countries versus high-income countries [2,3,4,5,6]. Several hypotheses regarding the disparity in vaccine efficacy between the two income groups exist [7,8,9,10]. Recent evidence suggests that gut bacteria also play a role in enteric viral infection and the immunogenicity of oral live attenuated viral vaccines [11,12,13]. However, the evidence is based on the overall composition of gut bacteria and not specific bacterial strains or bacterial compounds with immunomodulatory properties.

A successful rotavirus infection involves the attachment and entry of the virus into the enterocytes of the intestinal tract, where they undergo replication. Increasing evidence suggests that intestinal bacteria facilitate rotavirus infection. For instance, rotavirus was unable to infect mice treated with antibiotics compared to untreated ones [14]. While it is not entirely clear how intestinal bacteria facilitate rotavirus infection, the role of lipopolysaccharides (LPSs) expressed by Gram-negative bacteria have received increasing scrutiny. Binding to LPSs has been reported to improve thermostability and resistance to inactivation by the chemicals [15,16] of enteric viruses and promote their infection [17], perhaps by increasing the duration of time that the infectious virus is able to target intestinal cells. In addition, LPSs may also act as vaccine adjuvants, possibly through Toll-like receptor 4 (TLR4), indicating that LPS-rich bacteria may act as natural vaccine adjuvants [18] by mediating the immune response. The stimulation of TLR4 by LPSs induces the release of proinflammatory cytokines such as interleukin 8 (IL-8) [19]. Increased levels of IL-8 have been associated with rotavirus infection [20]. 

Bacterial LPS is a tripartite structure consisting of (1) the lipid A that docks the LPS to the outer membrane (2) core oligosaccharide, which maintain the integrity of the outer membrane; and (3) O antigen polysaccharide or O antigen that is connected to the core and made up of a polymer of repetitive oligosaccharide units exposed to and that interact with the external environment. The lipid A is the most conserved feature of LPS, which the host TLR4 recognises. However, there is a significant diversity in the LPS structure, even in lipid A, and consequently, different LPS structures have varying abilities to evoke the immune response [21]. For example, while Bacteroides LPS inhibits innate immune signalling and endotoxin tolerance, LPSs from Serratia, Klebsiella and others are reported to be toxigenic and potent immune activators [21]. Despite their importance in facilitating viral infection and priming the immune system, the role of these toxigenic LPSs and other bacteria-derived components known to bind enteric viruses, such as N-acetylglucosamine [16], in the paediatric response to live-attenuated, oral rotavirus vaccines has not been studied. This study hypothesises that the quantity of toxigenic LPS-rich bacteria in the gut may correlate with oral live attenuated rotavirus vaccine replication in infants. Low numbers of LPS-rich bacteria may result in a decreased amount of infectious vaccine virus available to infect epithelial cells, leading to decreased vaccine replication.

## 2. Methodology 

### 2.1. Study Design and Participants

This cross-section study used paired stool and saliva samples collected from infants attending Oukasie Healthcare Clinic, north of Pretoria, South Africa, for oral rotavirus vaccination using Rotarix (GlaxoSmithKline, Rixensart, Belgium) between 2019 and 2021. Stool and unstimulated saliva samples were collected from infants on the day of the first dose (6 weeks) and second dose (14 weeks) of vaccination prior to vaccine administration and 7 days after vaccination. Written informed consent was obtained from the parents prior to any study procedures. The vaccinated infants were divided into two groups based on vaccine virus shedding: rotavirus vaccine virus shedders and non-shedders. Only children weighing at least 2 kg and not on antibiotic treatment within 4 weeks prior to enrolment were included in the study. The study also used samples collected only at ages of 7 and 15 weeks. Demographic data such as sex, feeding type and birth delivery method were recorded on the infants’ case report forms (CRFs).

### 2.2. Ethics

Ethical clearance was granted by the Sefako Makgatho Health Sciences University Research and Ethics Committee (SMUREC), number SMUREC/M199/2020: PG. 

### 2.3. Specimen Collection and Storage

Stool samples were collected from the infant’s diapers into sterile containers and immediately placed on ice and taken to −20 °C freezers located at the clinic within 5 min after collection. Saliva samples were collected by first placing oral swabs in the mouth for 1 min, rubbing against the cheeks, and then placing them in sterile 15 mL tubes. All the specimens were transported frozen in cooler boxes containing ice blocks to the laboratory and immediately stored at −20 °C.

### 2.4. Stool Viral RNA and Genomic DNA Extraction

Viral RNA and genomic DNA were extracted from a 10% stool suspension (*w*/*v*) in sterile water using a Viral RNA Mini Extraction Kit (Qiagen, Hilden, Germany) and QIAamp Fast DNA Stool Mini Kit (Qiagen, Hilden, Germany), respectively, following the manufacturer’s instructions with few modifications [9]. The viral RNA and genomic DNA were eluted in 50 µL of nuclease-free water and 150 µL of elution buffer, respectively, and stored at −20 °C and −80 °C for short- and long-term storage, respectively.

### 2.5. Human and Bacterial mRNA Extraction

Human and bacterial mRNA were extracted from the saliva and stool samples, respectively, using a NucleoSpin RNA kit (Macherey-Nagel, Duren, Germany) following the manufacturer’s instructions with minor modifications [22]. Briefly, 200 µL of saliva was centrifuged for 15 min at 11,000 RCF at 4 °C, and human mRNA was extracted from the cell pellet. A 10% stool suspension (*w*/*v*) in sterile water was centrifuged, and bacterial mRNA was extracted from 150 µL of the supernatant. The human and bacterial mRNA were eluted in 50 µL RNA-free water and stored at −80 °C.

### 2.6. Detection of Rotavirus Vaccine Virus Shedding in Stool RNA Samples

The shedding of the live attenuated rotavirus vaccine virus in stool samples was detected via reverse-transcriptase real-time polymerase chain reaction (RT-qPCR) using a 2X Luna Universal qPCR Master Mix Kit (New England BioLabs, Ipswich, MA, USA) and primers and a probe targeting the NSP2 gene of the Rotarix vaccine strain [23]. The RT-qPCR was performed on a Bio-Rad CFX96 Real-Time System (Bio-Rad Laboratories, Hercules, CA, USA) as described previously (9). A value of a cycle threshold (Ct) < 40 was considered positive for NSP2 [24].

### 2.7. Primer and Probe Design for Selected Bacterial LPSs and N-acetylglucosamine Genes

Primers and probes for selected LPS-rich bacteria were designed from the genes *Waa*E (CP054780.1), *kdt*X (AP021873) and *Waa*L (NZ_CP015117.1) (Table 1), involved in the biosynthesis of LPS in *K. pneumoniae*, *S. marcescens* and *P. aeruginosa*, respectively, and *Glm*U for N-acetylglucosamine. The nucleotide sequences of the selected genes were aligned in MEGA X with sequences of similar genes from other bacterial species, and the primers and probes were designed from regions specific to the respective genes of interest. Primer specificity was confirmed through the conventional PCR of DNA of selected bacteria against DNA from other related bacteria.

### 2.8. Detection of the Selected LPS-Rich Bacteria in Stool DNA Samples

The presence and abundance of the selected LPS-rich bacteria in the stool DNA samples were determined via real-time qPCR using a Luna Universal Probe qPCR Mastermix kit (New England BioLabs) and primers and probes targeting the *Waa*E, *kdt*X, *Waa*L and *Glm*U genes. The 20 µL PCR reaction mix consisted of 1X Luna master mix, 400 nM of forward and reverse primers, 200 nM of the probes, 5 µL of DNA template and PCR grade water. Amplification was performed in a Bio-Rad CFX96 Real-Time System under the following conditions: 2 min of initial denaturation at 95 °C and 45 cycles of denaturation at 95 °C for 15 s and extension at 60 °C for 30 s. 

### 2.9. Bacterial Quantification 

Bacterial standard curves for the selected LPS-rich bacteria were generated from genomic DNA extracted from *K. pneumoniae* (ATCC: BAA-2146), *S. marcescens* (ATCC: 43862) and *P. aeruginosa* (ATCC: 27853) as described previously (24). The standard curve was generated through the PCR amplification of DNA isolated from 10-fold serial dilutions and the plotting of the Ct values of each dilution against their corresponding colony-forming units (CFU).

### 2.10. Bacterial and Host Gene Expression via Real-Time PCR

First-strand cDNA synthesis from RNA extracts was performed using a Tetro cDNA Synthesis kit (Bioline, Memphis, TN, USA,) following the manufacturer’s instructions. Briefly, the reaction mix consisted of 5 µL of mRNA, 1 µL of 100 µM random hexamers, 1 µL of 10 mM dNTP, 4 µL 5× RT Buffer, 1 µL of RiboSafe RNase Inhibitor and 7 µL of RNase-free water. The reverse transcription was performed on the GeneAmp PCR System 9700 (Applied Biosystems, Waltham, MA, USA) under the following conditions: 25 °C for 10 min, followed by 30 min of reverse transcription at 45 °C and the deactivation of the enzyme at 85 °C for 5 min. 

The expressions of the *Waa*E, *kdt*X, *Waa*L, *Glm*U, TLR-4 and IL-8 genes were measured via real-time PCR using a HOT FIREPol^®^ EvaGreen^®^ qPCR Mix Plus (ROX) (Solis BioDyne, Tartu, Estonia) and primers specific to the respective genes (Table 1). The 20 µL PCR reaction mixture consisted of 1× HOT FIREPol^®^ EvaGreen^®^ qPCR Mix Plus, 250 nM each of the forward and reverse primers and 5 µL of template and nuclease-free water. Amplification was performed on the Bio-Rad CFX96 Real-Time System using the following conditions: initial denaturation at 95 °C for 5 min, 45 cycles of denaturation at 95 °C for 10 s and extension at 60 °C for 15 s with a melt curve insertion (65 °C to 95 °C: Increment 0.5 °C for 0:05 s). Gene expression was normalised with the housekeeping gene GAPDH as previously described [26]. 

### 2.11. Statistical Analysis

Demographic data such as age, sex and feeding type for vaccine virus shedders and non-shedders were evaluated for statistical differences using a Chi-square test. The bacterial counts were converted into log form and analysed in GraphPad Prism 9.2.0 (GraphPad Software, San Diego, CA, USA). The distribution of the data was determined using Shapiro–Wilk or Kolmogorov–Smirnov tests. Descriptive statistics were represented by the median and interquartile range (IQR) and displayed as box plots. The Mann–Whitney U test was used to evaluate differences in bacterial abundance between the two study groups. Log counts (CFU/g) of selected LPS-rich bacteria were used as a continuous variable to test the association between bacterial abundance and vaccine virus shedding. Odds ratios (ORs) and 95% confidence intervals (CIs) were calculated using logistic regression. The contribution of potential confounding factors in predicting counts of selected LPS-rich bacteria was evaluated using multiple logistic regression. Fold-change gene expression was calculated using delta Ct as described by Schmittgen and Livak [26], where the normalised mean and standard deviation of the vaccine shedding infants were divided by those of the non-shedding infants. In all statistical tests, *p* ≤ 0.05 was deemed statistically significant.

## 3. Results

### 3.1. Demographics and Baseline Characteristics

A total of 78 vaccinated infants, 43 vaccine shedders and 35 non-shedders, were eligible for the study. All infants also received oral polio vaccine on the day of rotavirus vaccination. Except for the C-section delivery method, there were no significant differences in demographic data such as sex, age, feeding type and vaginal birth between the rotavirus vaccine shedders and non-shedders (Table 2).

### 3.2. Abundances of Selected LPS-Rich Bacteria and Bacterial N-acetylglucosamine in Stool Samples

Collectively, the bacterial counts of all selected LPS-rich bacteria were significantly higher in vaccine shedders (median 4.89 CFU/g, IQR 2.84) compared to non-vaccine shedders (median 3.13 CFU/g, IQR 2.74), *p* = 0.006 (Figure 1a). Individually, the counts of *S*. *marcescens* were also significantly higher in vaccine shedders (median 4.62 CFU/g, IQR 2.85) compared to their non-shedding counterparts (median 1.63 CFU/g, IQR 4.67, *p* = 0.0011 (Figure 1b). Although not statistically significant, the counts of *P. aeruginosa* in vaccine shedders (median 2.67 CFU/g, IQR 4.37) were higher than those in non-shedders (median 1.25, IQR 3.14), *p* = 0.0943 (Figure 1c). There were no statistically significant differences in the abundances for N-acetylglucosamine and *K. pneumoniae* between vaccine shedders [median 7.30 CFU/g, IQR 2.68 and median 7.58 CFU/g, IQR 2.28, respectively] and non-vaccine shedders [median 7.80 CFU/g, IQR 1.66 and median 7.25 CFU/g, IQR, 3.42, respectively] (Figure 1d,e).

When stratified according to age, the counts of *P. aeruguinosa* were significantly higher in 15-week-old vaccine shedders (median 3.067 CFU/g) than in their non-shedding counterparts (median 1.208), *p* = 0.0400 (Figure 1g). Conversely, the abundance of *S. marcescens* was significantly higher in 7-week-old vaccine shedders (median 4.74 CFU/g) compared to their non-shedding peers (median 2.31 CFU/g), *p* = 0.0014 (Figure 1f). There was no statistical difference in the abundance of *P. aeruguinosa* between 7-week-old vaccine shedders and non-shedders, *p* = 0.346. Neither was there any significant difference in the abundances of bacterial N-acetylglucosamine and *K. pneumoniea* in both age groups between the two study populations.

### 3.3. Expression of Selected Bacterial LPS and N-acetylglucosamine Genes in Vaccine Shedders versus Non-Shedders

Unlike the abundance, the average expressions of all selected LPS genes were similar between the two study groups (37.21 ± 16.43 vs. 39.69 ± 15.32). Individually, the expressions of the three selected genes were variable between the two groups. Contrary to the abundance, there was a three-fold reduction in the expression of *kdt*X (S. *marcescens*) in vaccine shedders (5.75 ± 1.64) compared to non-shedders (18.73 ± 5.15) (Figure 2b). On the other hand, the expression of *Waa*E (*K. pneumonia*) increased two-fold in vaccine shedders (20.60 ± 6.84) compared to non-shedders (13.23 ± 4.75) (Figure 2c). There was no difference in the expression levels of gene *Glm*U (N-acetylglucosamine) between vaccine shedders (85.28 ± 39.41) and non-shedders (87.11 ± 36.06) (Figure 1a). 

### 3.4. Levels of TLR4 and IL-8 Gene Expressions 

The salivary and stool expression levels of genes coding for TLR4 and IL-8 (salivary) were compared between vaccine shedders and non-shedders. As shown in Figure 3, the expression levels of the two genes corresponded with the abundance of selected LPS-rich bacteria. The mean expression level of the TLR4 gene in stool samples was increased two-fold in vaccine shedders compared to non-shedders (Figure 3b). In saliva, the mean expression levels of the TLR4 and IL-8 genes increased six- and four-fold in vaccine shedders (51.98 ± 36.42 and 9.64 ± 6.82, respectively) compared to their non-shedding counterparts (22.52 ± 11.75 and 6.67 ± 2.37, respectively) (Figure 3a,c).

### 3.5. Association between Abundances of Selected LPS-Rich Bacteria and Vaccine Shedding

The average counts of all selected LPS-rich bacteria (Ct values) and shedding status (yes or no) were used to evaluate their association with rotavirus vaccine shedding in the stool samples of the study infants. Logistical regression analysis indicated that infants with high counts of selected LPS-rich bacteria were 1.5 times likely to shed the vaccine in stool samples (95% CI = 1.1–1.89), and there was a significant association between the counts of LPS-rich bacteria and vaccine shedding, *p* = 0.007. The vaccine shedding (Ct values) was used as a quantitative variable to evaluate its association with the abundance of LPS, and the results (Figure 4) validated the association between the abundance of bacterial LPS and vaccine shedding (yes or no) variable. 

Potential confounding factors such as sex, the birth delivery method and the infant’s feeding type were also evaluated for their contributions in predicting the counts of the selected LPS-rich bacteria. The three potential compounding factors did not contribute in predicting the counts of the selected LPS-rich bacteria between the two study groups 

## 4. Discussion

Diversity in the LPS structure among bacterial species and strains presented an obstacle in designing primers that can target all LPSs in stool samples because of a lack of common conserved regions in LPS genes. This diversity in LPS structures, especially the lipid A, brings with it a varying ability to trigger the immune response. For example, LPS from some bacteria, e.g., Bacteroides, has been reported to dampen host immune responses, while others, e.g., some Proteobacteria, are toxigenic and potent immune activators. As such, we selected *Serratia*, *Klebsiella* and *Pseudomonas* not only as representatives of toxigenic bacterial LPS that strongly induce immune response [21], but also for being LPS rich. Although they have provided little about the mucosal immune response, most studies have measured the levels of serum IgM or IgG to assess the rotavirus vaccine response [12,13]. Due to the unavailability of serum, the current study used vaccine virus shedding in stool samples as a proxy for vaccine response as it signals the vaccine virus replication that is required for mucosal immunity and vaccine effectiveness [27]. 

This study observed a significantly greater abundance of the selected LPS-rich bacteria in the stool samples of infants who were vaccine shedders compared to their non-shedding counterparts. This is the first study to report such significant differences in the abundance of LPS-rich bacteria between these two study groups. Significant differences in the composition of the gut bacteria between vaccine responders and non-responders have previously been reported [12,13]. However, those reports were based on general changes in gut bacteria and not specific bacterial strains or bacteria-derived molecules with immunomodulatory capabilities that can influence viral infection, as reported in the current study. Bacterial LPSs can promote viral infection through several mechanisms, including facilitating the attachment of the virus to the target cells, and the observations in the current study suggest that harbouring a high abundance of LPS-rich bacteria may enhance vaccine virus attachment and replication in the target epithelial cells, leading to better vaccine responses. Indeed, both the linear and logistic regression analyses of the counts of the selected LPS-rich bacteria between vaccine shedders and non-shedders indicated that possessing higher abundances of LPS-rich bacteria increases the chance of the vaccine virus to be shed in a stool sample.

Individually, the abundance of *S. marcescens* was significantly higher in the stool samples of vaccine shedders compared to non-shedders. Although not targeting *S. marcescens* individually, a similar study found Bacilli, to which *S. marcescens* belongs, was significantly higher in vaccine responders compared to non-responders [13]. *S. marcescens*, like other Gram-negative bacilli such as *Escherichia coli* and *Klebsiella*, possesses toxigenic LPS, which triggers an inflammatory response [21]. It is possible that the high abundance of LPS-rich *S. marcescens* in vaccine shedders could have primed the immune system or acted as a natural vaccine adjuvant, subsequently improving vaccine replication. Thus, LPS could be used as a vaccine adjuvant. However, its strong biological activity can contribute to vaccine reactogenicity, as such a modification of its structure will be required to allow the triggering of a proper immune response while at the same time lessening its toxic properties. 

The abundance of *S. marcescens* was significantly higher in 7-week-old vaccine shedders compared to their non-shedding counterparts, while that of *P. aeruguinosa* was significantly higher in 15-week-old vaccine shedders than in 15-week-old non-shedders. The composition of gut bacteria during infancy is dynamic and unstable [28], and the findings in the current study suggest that the increased abundance of any of the LPS-rich bacteria at the time of rotavirus vaccination would be a susceptibility factor for vaccine take.

The average total expression of selected LPS genes, unlike the abundance, was not different between vaccine shedders and non-shedders, suggesting that the abundance of bacteria does not necessarily correspond with the expression level of the target gene. The qPCR assay employed to quantify the selected LPS-rich bacteria estimates only the copy numbers of the targeted genes. It does not indicate the number of viable bacteria. Nonetheless, the promotion of enteric viral infection by bacterial LPS is not limited to viable bacteria, bacterial LPS compounds on their own can also enhance viral infection sufficiently [16].

LPS signalling for enteric bacteria occurs through TLR4, which results in the induction and release of a cascade of proinflamatory cytokines. The current study found that the expression of TLR4 in stool samples was consistent with the abundance of selected bacterial LPS. This is consistent with studies on mice that showed that the LPS activation of TLR4 genes proceeds in a dose-dependent manner [29]. The current study also observed that the abundances of all selected LPS-rich bacteria corresponded with the levels of the salivary TLR4 gene expressions in the vaccine shedders and non-shedders, suggesting that the amount of LPS is positively related to the TLR4 gene expression. This is consistent with a recent study that indicated that the immune mucosal response to RV infection is not confined to the intestinal mucosa but to oral mucosa as well [30]. In addition, TLRs, including TLR4, have been shown to be expressed along all of the alimentary canal [31]. 

The activation of TLR4 induces the release of proinflamatory cytokines such as IL-8 [32,33]. This is consistent with the observation in the current study, in which high levels of TLR4 gene expression corresponded with those of IL-8. However, the release of IL-8 has also been shown to be induced by rotavirus infection [34]. Nevertheless, since both study groups were exposed to the rotavirus vaccine, it is possible that an increase in the expression of IL-8 could be due to the elevated activation of TLR4 by the LPSs. This is consistent with increased levels of the IL-8 expression that were also observed in the rotavirus infection in the presence of LPS-rich *Salmonella* [20]. In addition, an increase in the IL-8 gene expression could have contributed to the improved replication and shedding of the vaccine virus in stool samples. This would be consistent with findings from a study that observed a positive association between the expression of IL-8 with the improved replication of rotavirus infection [32]. Further alluding to the involvement of IL-8 in LPS-associated viral replication, high levels of IL-8 have been reported in childhood viral gastroenteritis [20]. 

The study had several limitations. First, the number of samples used was small and required a guarded interpretation of the results. Further studies with larger sample sizes are required to validate the findings. Second, vaccine response was measured based on vaccine shedding in stool samples. However, studies have shown that some infants who do not shed the vaccine in stool samples have a strong serological response. As such, some samples may have been classified as a non-vaccine response based on the shedding when they were responders serologically. The study was conducted in a predominantly black township, where the standard of living is poor, meaning that the results may not be generalisable to other settings. Another limitation is the lack of an unbiased metagenomic analysis of the microbiome in the infant stool samples that would have been compared between rotavirus vaccine shedders and non-shedders to strengthen the findings of the study. Lastly, the study was also not able to perform a serological quantification of the cytokines to further validate the findings.

In summary, this study has shown significant differences in the abundance of selected LPS-rich bacteria as well as the expressions of the TLR4 and IL-8 genes between infants who shed the rotavirus vaccine in stool samples compared to those of non-shedders. There was a positive association between the abundances of selected LPS-rich bacteria and live attenuated rotavirus vaccine shedding, suggesting that bacterial LPS could play a role in rotavirus vaccine take.

## Figures and Tables

**Figure 1 vaccines-12-00273-f001:**
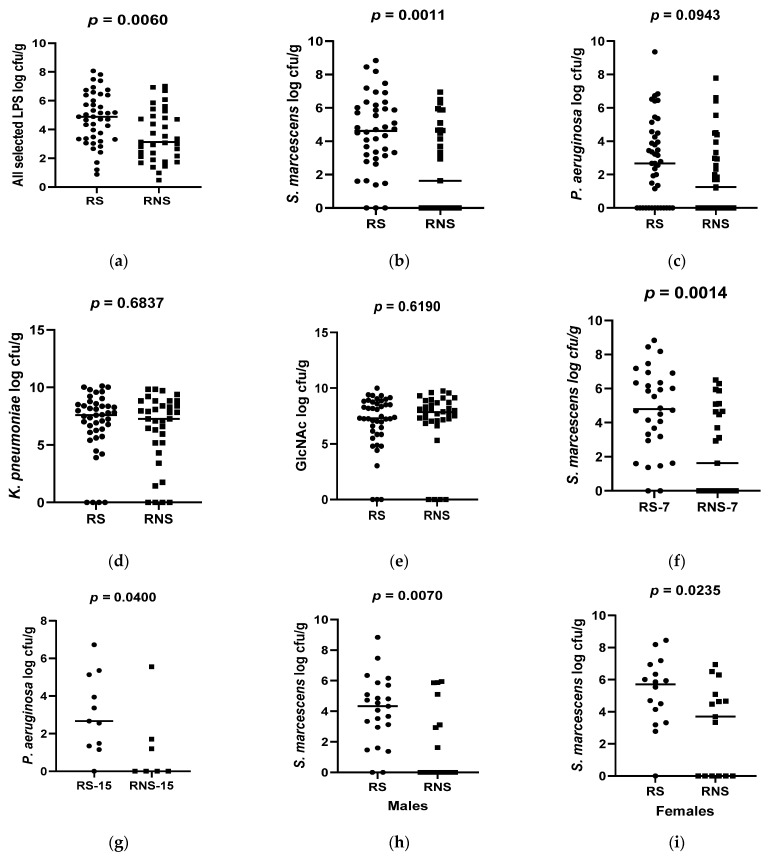
Box plots showing abundances of selected LPS-rich bacteria and N-acetylglucosamine 7 between rotavirus vaccine shedders (RS) and non-shedders (RNS): (**a**) average of all selected LPS-bacteria (**b**) *S. marcescens* (**c**) *P. aeruguinosa* (**d**) *K. pneumonia* (**e**) N-acetylglucosamine (**f**) *S. marcescens* at 7 weeks of age (**g**) *P. aeruguinosa* at 15 weeks of age (**h**) *S. marcescens* in males (**i**) *S. marcescens* in females.

**Figure 2 vaccines-12-00273-f002:**
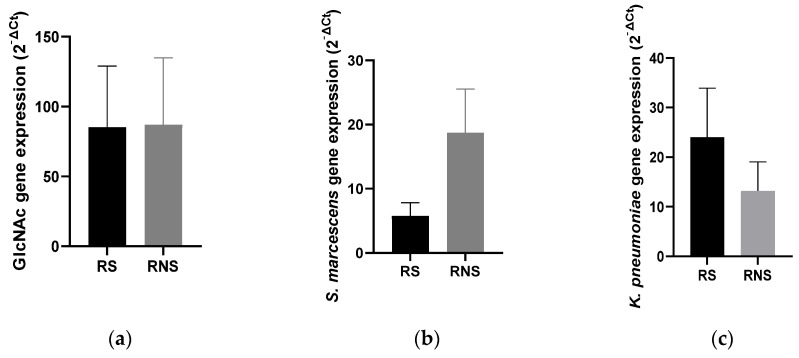
Bar charts showing average expression levels of genes coding for LPS and N-acetylglucosamine in selected LPS-rich bacteria between rotavirus vaccine shedders and their non-shedding counterparts: (**a**) N-acetylglucosamine (**b**) *kdt*X (*S. marcescens*) (**c**) *Waa*E (*K. pneumonia*).

**Figure 3 vaccines-12-00273-f003:**
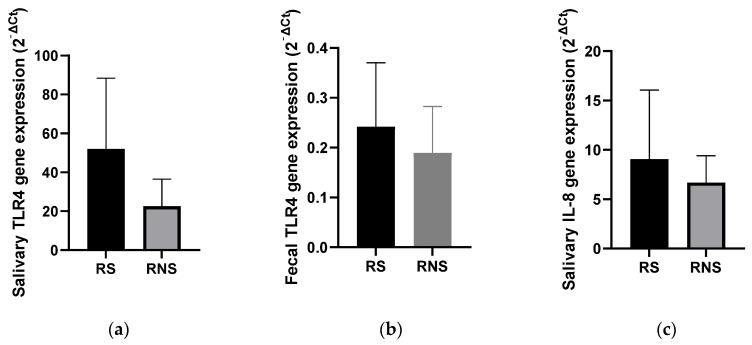
Bar charts showing expression levels of genes coding for TLR4 and IL-8 in stool saliva samples of rotavirus vaccine shedding (RS) and non-shedding (RNS) infants: (**a**) TLR4 expression in saliva (**b**) TLR4 expression in fecal samples (**c**) IL-8 expression in saliva samples.

**Figure 4 vaccines-12-00273-f004:**
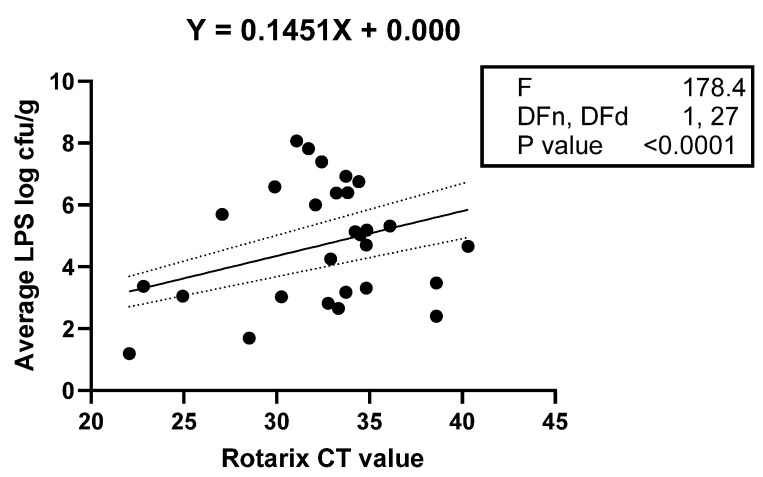
Linear regression showing a statistically significant linear relationship between Rotarix vaccine stool shedding (X and Y), as indicated by the low *p*-value for the F-test. The dotted lines indicate the upper and lower bounds of the 95% confidence interval for the re-gression line represented by solid line.

**Table 1 vaccines-12-00273-t001:** Primers and probes used in the study.

Bacteria/Compound	Primer Name and Sequences	Target Gene	Reference
*P. aeruguinosa*	WaaL F-CCAGATCAGCGAGCATCCAT	*Waa*L	This study
	WaaL R-CGAAAAGCACACCCAGTTCG		
	Waal P-Texas red-CGGCTACGATCATCCGAT-BHQ-2		
*S. marcescens*	Waa F-TCGACGGTAAACAGGGGTTG	*kdt*X	This study
	Waa R-CGAACGTCCCGGGATAGATG		
	Waa P-Hex-TAGCGGTGGTCAACGCGCAATATA		
*K. pneumoniae*	WaaEF-TCGTTATAGCGGTAACGGGC	*Waa*E	This study
	WaaER-TCGCCCGCCGTAACTATTTT		
	WaaEP-Hex-ATACCAACCGCTGTGGCGCATAAA-BHQ-1		
N-acetylglucosamine	GlmU-F-GTGATGTAGTATTCGCCCTGAG	*Glm*U	This study
	GlmU-R-AAGATGCCACCGACGAGCAG		
	GlmU-P FAM-TTGTTGGTCAGCTTCGCCAG-BHQ-1		
Interleukin 8	IL-8 F–CACCGGAAGGAACCATCTCACT	IL-8	This study
	IL-8 R–ACCTTCACACAGAGCTGCAGA		
Toll-like receptor 4	TLR4 F–GATTGCTCAGACCTGGCAGTT	TLR4	This study
	TLR4 R-GTCCTCCCACTCCAGGTAAGT		
Glyceraldehyde 3-phosphate dehydrogenase	Gapdh-F–CAAGGTCATCCATGACAACTTTG	Gapdh	[25]
	Gapdh-R–GTCCACCACCCTGTTGCTGTAG		

**Table 2 vaccines-12-00273-t002:** Demographic data and other baseline characteristics of the study participants.

Characteristic	Vaccine Shedders (*n* = 43), *n* (%)	Vaccine Non-Shedders (*n* = 35), *n* (%)	*p* Value
Sex **Male**	25 (58.14)	19 (54.29)	0.8198
Female	18 (41.86)	16 (45.71)	
Age			
7 weeks	32 (74.42)	27 (77.14)	>0.9999
15 weeks	11 (25.58)	8 (22.86)	
Birth method			
Natural birth	27 (62.79)	27 (77.14)	0.2204
Caesarean	16 (37.21)	8 (22.86)	
Feeding type			
Breast milk	37 (86.05)	30 (85.71)	>0.9999
Formula	6 (13.95)	5 (14.29)	

## Data Availability

The data presented in this study are available on request from the corresponding author.

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
