# Peer review of "Abundance of Selected Lipopolysaccharide-Rich Bacteria and Levels of Toll-like Receptor 4 and Interleukin 8 Expression Are Significantly Associated with Live Attenuated Rotavirus Vaccine Shedding among South African Infants"

_vaccines, 2024, doi:10.3390/vaccines12030273_

Round 1
Reviewer 1 Report
Comments and Suggestions for Authors
Based on the reported evidence that bacterial lipopolysaccharides (LPS) have been shown to promote enteric viral infections, the authors want to test the hypothesis that elevated levels of bacterial LPS improves oral rotavirus vaccine replication in South African infants. 78 infants (7w, 15w) a week after rotavirus vaccination were enrolled to collect stool and saliva samples for study. 78infants were grouped into vaccine virus shedders (n = 43) and non-shedders (n = 35). Abundance of LPS-rich bacteria, including Serratia marcescens, Pseudomonas aeruguinosa and Klebsiella pneumonia, as well as LPS and IL-8 gene expression in samples were assessed by RT-QPCR, and found that 1) the abundance of selected LPS-rich bacteria was significantly higher in vaccine shedders (median log 4.89 CFU/g, IQR 2.84) compared to non-shedders (median log 3.13 CFU/g, IQR 2.74), p = 0.006. 2) TLR4 and IL-8 gene expression was increased four and two-fold, respectively, in vaccine shedders versus non-shedders, but no difference was observed in bacterial LPS expression, p = 0.09. 3) Regression analysis indicated a significant association between abundance of selected LPS-rich bacteria and vaccine virus shedding (Odds ratio 1.5, 95% 23 CI = 1.10 – 1.89), p = 0.002. All the data suggest that harbouring higher counts of LPS-rich bacteria can increase oral rotavirus vaccine take in in infants. This study is meaningful.
Major concern
1. In the results part, the figure name should insert to the corresponding description,
2. The Y-axis illustration should be accurate,
3. Does the TLR4 expression correlate with LPS rich-bacteria?
4. Line 45-49, it is reported that virus can bind LPS to improve viral thermostability and resistance to inactivation by chemicals, indicating that all kinds of viruses can bind LPS, and how? What is the mechanism underlying the LPS acting as vaccine adjuvants? isn't it activating the signal pathways (LPS=TLR4) to regulate immune response? LPS as adjuvants or influencing the viral replication, depending on the LPS-TLR4 signal pathway activation.
5. IL-8 maybe is just the marker of LPS signal pathway activation, how it exerts its effect in improvement of vaccine lacks data to support it.
6. Based on the authors’ discovery, could the LPS be the adjuvant for rotavirus? Is it safe for the infants? How to control the usage dose?
7. Is the shedding or non-shedding the most convenient, cheapest method to evaluate the efficacy of rotavirus vaccine? What is the most reliable evaluation parameters currently? There is any comparison between the methods used in this study and the most reliable method?
Minor concerns
1. Introduce the character of the rotavirus vaccine, which is live virus vaccine.
2. Figure 2 and Figure 3, Despite there are multiple change between the two groups. does it exist significant change?
3. line 183,185, 198, high should be higher.
Comments on the Quality of English LanguageThe writing of this manuscript should be improved.
Author Response
Reviewer 1
Based on the reported evidence that bacterial lipopolysaccharides (LPS) have been shown to promote enteric viral infections, the authors want to test the hypothesis that elevated levels of bacterial LPS improves oral rotavirus vaccine replication in South African infants. 78 infants (7w, 15w) a week after rotavirus vaccination were enrolled to collect stool and saliva samples for study. 78infants were grouped into vaccine virus shedders (n = 43) and non-shedders (n = 35). Abundance of LPS-rich bacteria, including Serratia marcescens, Pseudomonas aeruguinosa and Klebsiella pneumonia, as well as LPS and IL-8 gene expression in samples were assessed by RT-QPCR, and found that 1) the abundance of selected LPS-rich bacteria was significantly higher in vaccine shedders (median log 4.89 CFU/g, IQR 2.84) compared to non-shedders (median log 3.13 CFU/g, IQR 2.74), p = 0.006. 2) TLR4 and IL-8 gene expression was increased four and two-fold, respectively, in vaccine shedders versus non-shedders, but no difference was observed in bacterial LPS expression, p = 0.09. 3) Regression analysis indicated a significant association between abundance of selected LPS-rich bacteria and vaccine virus shedding (Odds ratio 1.5, 95% 23 CI = 1.10 – 1.89), p = 0.002. All the data suggest that harbouring higher counts of LPS-rich bacteria can increase oral rotavirus vaccine take in in infants. This study is meaningful.
Major concern
- In the results part, the figure name should insert to the corresponding description,
Agreed. The figure names have been inserted
- The Y-axis illustration should be accurate,
Agreed, the Y axis of some of the figures have been revised
- Does the TLR4 expression correlate with LPS rich-bacteria?
Since the qPCR used to quantify the abundance of LPS-rich bacteria only provides copy numbers of the selected genes but not whether the bacteria was viable or not, we decided not to evaluate the correlation between TLR4 and abundance of LPS rich-bacteria. However, the expression levels of TLR4 corresponded with abundance of selected LPS-rich bacteria as mentioned in results sections
- Line 45-49, it is reported that virus can bind LPS to improve viral thermostability and resistance to inactivation by chemicals, indicating that all kinds of viruses can bind LPS, and how? What is the mechanism underlying the LPS acting as vaccine adjuvants? isn't it activating the signal pathways (LPS=TLR4) to regulate immune response? LPS as adjuvants or influencing the viral replication, depending on the LPS-TLR4 signal pathway activation.
Line 45-49: First of all we have revised the statement to mean that theromostability and resistance to chemicals has been shown in enteric viruses not all viruses. We have also modified the statement about LPS as an adjuvant to indicate that this is achieved through TLR4 by mediating the immune response.
- IL-8 maybe is just the marker of LPS signal pathway activation, how it exerts its effect in improvement of vaccine lacks data to support it.
We agreed that more research need to be done understand how IL-8 improves vaccine take. However, some studies have shown that rotavirus infection induces IL-8 and colleration between IL-8 and rotavirus infection as mentioned in the discussion section.
- Based on the authors’ discovery, could the LPS be the adjuvant for rotavirus? Is it safe for the infants? How to control the usage dose?
The LPS could be used as vaccine adjuvant. However, its strong biological activity can contribute to vaccine reactogenicity, thus modification of its structure will be required to allow triggering of a proper immune response while at the same time lessening its toxic properties. This has been added in the discussion section
- Is the shedding or non-shedding the most convenient, cheapest method to evaluate the efficacy of rotavirus vaccine? What is the most reliable evaluation parameters currently? There is any comparison between the methods used in this study and the most reliable method?
Other similar studies have used 3 to 4-fold increase in serum IgG or IgM as an indicator of vaccine response. However, the study was unable to draw blood from infants due to the reluctance of parents to do so. The other option was to measure stool IgG. However, stool IgG degrades very quickly and would need liquid nitrogen to immediately freeze the sample at the time of collection. We did not have the capability to store liquid nitrogen the study clinic. We opted for shedding as a proxy for vaccine response since stool vaccine shedding signals vaccine virus replication that is required for mucosal immunity and vaccine effectiveness
Minor concerns
- Introduce the character of the rotavirus vaccine, which is live virus vaccine.
Agreed: The word ‘live attenuated’ has been inserted in the title and other sections in the text to describe the nature of the vaccine.
- Figure 2 and Figure 3, Despite there are multiple change between the two groups. does it exist significant change?
We do not understand the question
- line 183,185, 198, high should be higher.
Agreed. The word has been changed to higher
Comments on the Quality of English Language
The writing of this manuscript should be improved.
We have sent the manuscript to several experience writers for review and the current version took in all their suggestions.
Reviewer 2 Report
Comments and Suggestions for Authors
Poor response to oral rotavirus (RV vaccine particularly human RV vaccine Rotarix, is a well-known problem in developing countries, including Africa. The non-responsiveness cannot be compensated by administration of multiple doses. As pointed out by the authors, multiple explanations to this observation have been put forward, but none of them are entirely satisfactory. Furthermore, there is no remedy.
To that end the present study is a potentially valuable contribution. The study shows that there is a correlation between the presence of certain LPS-rich enteric bacteria (Serratia, Klebsiella, Pseudomonas) in the gut and RV vaccine uptake as measured by vaccine virus shedding. The positive effect of these bacteria in gut microbiota has been shown previously, but the present study provides an easier method for such studies by measuring LPS expression using RT-q PCR. In addition, the authors also present some kind of explanation for the mechanism of LPS effect via increased expression of TLR-4 and IL-8 genes.
Despite its merits the present study should be described as a preliminary report with several (but not critical) weaknesses. These are related to other points but the authors’ expertise.
1. The division of infants to shedders and non-shedders is artificial (although helpful and simple for the purposes of the study). A more appropriate way would be quantitation (by RT-PCR) of the vaccine virus shedding and then making correlations with the measurements of LPS gene expression
2. An even more appropriate way would be to study the immune responses after vaccination, because this is the real indicator of vaccine uptake
3. Interference by OPV is one factor in reduced RV vaccine uptake in South Africa. Information on concomitant OPV administration should be given
As a separate point, Table 2 could (should) be deleted. The only relevant information is about breast-feeding, and that can be given in the text.
Author Response
Poor response to oral rotavirus (RV vaccine particularly human RV vaccine Rotarix, is a well-known problem in developing countries, including Africa. The non-responsiveness cannot be compensated by administration of multiple doses. As pointed out by the authors, multiple explanations to this observation have been put forward, but none of them are entirely satisfactory. Furthermore, there is no remedy.
To that end the present study is a potentially valuable contribution. The study shows that there is a correlation between the presence of certain LPS-rich enteric bacteria (Serratia, Klebsiella, Pseudomonas) in the gut and RV vaccine uptake as measured by vaccine virus shedding. The positive effect of these bacteria in gut microbiota has been shown previously, but the present study provides an easier method for such studies by measuring LPS expression using RT-q PCR. In addition, the authors also present some kind of explanation for the mechanism of LPS effect via increased expression of TLR-4 and IL-8 genes.
Despite its merits the present study should be described as a preliminary report with several (but not critical) weaknesses. These are related to other points but the authors’ expertise.
- The division of infants to shedders and non-shedders is artificial (although helpful and simple for the purposes of the study). A more appropriate way would be quantitation (by RT-PCR) of the vaccine virus shedding and then making correlations with the measurements of LPS gene expression
Agreed. The shedding or not shedding was based on CT values of vaccine virus in stool samples and there was a positive correlation between shedding ct values and LPS gene expression i.e. for Serratia
- An even more appropriate way would be to study the immune responses after vaccination, because this is the real indicator of vaccine uptake
We agree with the reviewer that measuring immune response would be appropriate. However, we had a challenge in collecting blood from the babies as mothers were unwilling for blood to be drawn from their babies hence the use of vaccine shedding as a proxy of vaccine response since stool vaccine shedding signals vaccine virus replication that is required for mucosal immunity and vaccine effectiveness.
- Interference by OPV is one factor in reduced RV vaccine uptake in South Africa. Information on concomitant OPV administration should be given
Agreed. The information has been given in results section under demographics and other baseline characteristics, line 176-177
As a separate point, Table 2 could (should) be deleted. The only relevant information is about breast-feeding, and that can be given in the text.
We agree with the reviewer but we were trying to pre-empt other reviewers asking for demographic data.
Reviewer 3 Report
Comments and Suggestions for Authors
The authors sampled stool and saliva from 78 infants in South Africa vaccinated against rotavirus. The methodology and results were clearly and concisely presented and demonstrated a correlation between the presence of select LPS-rich bacteria and vaccine virus shedding. Given the previously established correlations between LPS and enhancement of enteric virus stability and infectivity, examination of the relationship between LPS rich bacteria and efficacy of rotavirus vaccines in warranted. There was a major concern with how the authors state the study was performed compared to the samples that were analyzed. This likely reflects the need for more detail in the Study design section of the methods. There were also some minor issues that should be corrected to improve the rigor or clarity of the manuscript.
Major concerns
*Line 68. Study Design. The authors state that stool was collected prior to vaccine administration. How was shedding of the vaccine virus detected if samples were only taken prior to vaccine administration? Please clearly state the other sampling points.
Line 77-78: In addition, the methods state that the study only used samples collected at age 7 weeks, but data within the manuscript compares shedding and LPS rich bacterium levels at 7 weeks and 15 weeks. Please correct the methods so they accurately reflect what collection procedures were performed or explain the discrepancy.
Minor concerns
*Line 46: It should be revised to clarify that binding to LPS has been shown to enhance thermal stability and promote viral infection of enteric viruses. Since rotavirus is the only virus mentioned prior to this statement, the statement implies that these LPS associated activities have been shown for rota, when the articles cited are for research done on poliovirus. The other reference cited is for a review article about rotavirus and not a primary research article demonstrating stability nor viral infection.
Fig 3B. The image is stretched and thus does not look consistent with panels A and C. This should be corrected.
Line 245: The font size of the text on this line appears different than the rest of the manuscript. It is not clear if this is a file conversion error or something that can be corrected by the authors.
Line 259: Authors should revise to say “LPS can promote viral infection”. LPS does not universally promote infection of viruses nor has it been shown to promote infection for all enteric viruses (e.g. norovirus).
Line 280: Authors state their study suggests that an “increased abundance of any LPS-rich bacteria at the time of rotavirus vaccination would be a risk factor for vaccine response.” Why is it a risk factor? The authors should further explain this statement.
Author Response
The authors sampled stool and saliva from 78 infants in South Africa vaccinated against rotavirus. The methodology and results were clearly and concisely presented and demonstrated a correlation between the presence of select LPS-rich bacteria and vaccine virus shedding. Given the previously established correlations between LPS and enhancement of enteric virus stability and infectivity, examination of the relationship between LPS rich bacteria and efficacy of rotavirus vaccines in warranted. There was a major concern with how the authors state the study was performed compared to the samples that were analyzed. This likely reflects the need for more detail in the Study design section of the methods. There were also some minor issues that should be corrected to improve the rigor or clarity of the manuscript.
Major concerns
*Line 68. Study Design. The authors state that stool was collected prior to vaccine administration. How was shedding of the vaccine virus detected if samples were only taken prior to vaccine administration? Please clearly state the other sampling points.
Agreed and the statement about sample collection points has been modified to also include 7 days post vaccination.
Line 77-78: In addition, the methods state that the study only used samples collected at age 7 weeks, but data within the manuscript compares shedding and LPS rich bacterium levels at 7 weeks and 15 weeks. Please correct the methods so they accurately reflect what collection procedures were performed or explain the discrepancy.
Agreed: The statement has been modified to indicate that samples used were collected at 7 and 15 weeks.
Minor concerns
*Line 46: It should be revised to clarify that binding to LPS has been shown to enhance thermal stability and promote viral infection of enteric viruses. Since rotavirus is the only virus mentioned prior to this statement, the statement implies that these LPS associated activities have been shown for rota, when the articles cited are for research done on poliovirus. The other reference cited is for a review article about rotavirus and not a primary research article demonstrating stability nor viral infection.
Agreed and the statement has been revised. We have also removed the reference cited and replaced with an appropriate one.
Fig 3B. The image is stretched and thus does not look consistent with panels A and C. This should be corrected.
Agreed. The figure has been corrected
Line 245: The font size of the text on this line appears different than the rest of the manuscript. It is not clear if this is a file conversion error or something that can be corrected by the authors.
The font on the said line appear to us the same as the rest of the text.
Line 259: Authors should revise to say “LPS can promote viral infection”. LPS does not universally promote infection of viruses nor has it been shown to promote infection for all enteric viruses (e.g. norovirus).
Agreed and the sentence has been modified and now reads ‘LPS can promote viral infection’..
Line 280: Authors state their study suggests that an “increased abundance of any LPS-rich bacteria at the time of rotavirus vaccination would be a risk factor for vaccine response.” Why is it a risk factor? The authors should further explain this statement.
We were trying to say that infants with high abundance of the selected bacteria would be more susceptible to vaccine take. ‘Risk’ was not a good choice of word and have replaced it with susceptibility to vaccine take in the text.
Reviewer 4 Report
Comments and Suggestions for Authors
This study investigates the impact of elevated levels of bacterial lipopolysaccharides (LPS) on the replication of oral rotavirus vaccine in South African infants. The rationale behind the research stems from the known association between bacterial LPS and the promotion of enteric viral infections. The study conducted stool sample analysis from infants one week after rotavirus vaccination to distinguish between vaccine virus shedders and non-shedders. Quantitative real-time PCR was employed to assess the presence of selected LPS-rich bacteria, such as Serratia marcescens, Pseudomonas aeruginosa, and Klebsiella pneumonia. Additionally, the study measured the gene expression of bacterial LPS, host Toll-like Receptor 4 (TLR4), and Interleukin-8 (IL-8). The key findings reveal a significant increase in the abundance of selected LPS-rich bacteria in vaccine shedders compared to non-shedders, suggesting a potential link between bacterial LPS and the replication of the oral rotavirus vaccine.
Regression analysis further indicated a significant association between the abundance of selected LPS-rich bacteria and vaccine virus shedding, with an Odds ratio of 1.5 and a 95% confidence interval of 1.10 – 1.89 (p = 0.002). This suggests that higher counts of LPS-rich bacteria may enhance the efficacy of oral rotavirus vaccine in infants. The study provides valuable insights into the potential role of bacterial LPS in influencing the replication of oral rotavirus vaccine, emphasizing the importance of understanding the interplay between gut bacteria and vaccine responses for effective immunization strategies.
In the study, there is a lack of serological quantification of cytokines that could confirm the data.
Additionally, bacterial isolation from fecal samples is missing to validate the information.
Further clarification is needed on how vaccine shedding is associated with a better response in vaccinated subjects.
Moreover, it is necessary to provide a more detailed explanation in the text regarding the rationale for choosing these bacteria.
Author Response
This study investigates the impact of elevated levels of bacterial lipopolysaccharides (LPS) on the replication of oral rotavirus vaccine in South African infants. The rationale behind the research stems from the known association between bacterial LPS and the promotion of enteric viral infections. The study conducted stool sample analysis from infants one week after rotavirus vaccination to distinguish between vaccine virus shedders and non-shedders. Quantitative real-time PCR was employed to assess the presence of selected LPS-rich bacteria, such as Serratia marcescens, Pseudomonas aeruginosa, and Klebsiella pneumonia. Additionally, the study measured the gene expression of bacterial LPS, host Toll-like Receptor 4 (TLR4), and Interleukin-8 (IL-8). The key findings reveal a significant increase in the abundance of selected LPS-rich bacteria in vaccine shedders compared to non-shedders, suggesting a potential link between bacterial LPS and the replication of the oral rotavirus vaccine.
Regression analysis further indicated a significant association between the abundance of selected LPS-rich bacteria and vaccine virus shedding, with an Odds ratio of 1.5 and a 95% confidence interval of 1.10 – 1.89 (p = 0.002). This suggests that higher counts of LPS-rich bacteria may enhance the efficacy of oral rotavirus vaccine in infants. The study provides valuable insights into the potential role of bacterial LPS in influencing the replication of oral rotavirus vaccine, emphasizing the importance of understanding the interplay between gut bacteria and vaccine responses for effective immunization strategies.
In the study, there is a lack of serological quantification of cytokines that could confirm the data.
Agreed and we have mentioned this as one of the limitations of the study
Additionally, bacterial isolation from fecal samples is missing to validate the information.
Bacterial isolation is done through conventional culture based media which only detects about 5% of bacteria in the gut and the rest are uncultivable. So using bacterial isolation would not give a true representative of the abundance of the selected bacteria. That’s why we performed LPS gene expression instead.
Further clarification is needed on how vaccine shedding is associated with a better response in vaccinated subjects.
Agreed. We have provided that information at the beginning of the discussion section.
Moreover, it is necessary to provide a more detailed explanation in the text regarding the rationale for choosing these bacteria.
We have tried to expand on the reasons for selecting the three bacteria in the beginning of the discussion section
Round 2
Reviewer 1 Report
Comments and Suggestions for Authors
All the questions have been addressed, it looks better.
Reviewer 4 Report
Comments and Suggestions for Authors
The changes made are acceptable, and the manuscript can be considered complete.